# Vaccination of SARS-CoV-2-infected individuals expands a broad range of clonally diverse affinity-matured B cell lineages

Mark Chernyshev [1,6], Mrunal Sakharkar[2,6], Ruth I. Connor [3], Haley L. Dugan[2], Daniel J. Sheward [1], C. G. Rappazzo[2], Aron Stålmarck [1], Mattias N. E. Forsell [4], Peter F. Wright[3], Martin Corcoran[1], Ben Murrell [1,6], Laura M. Walker[2,5,6] ✉ & Gunilla B. Karlsson Hedestam[1,6] ✉

Vaccination of SARS-CoV-2 convalescent individuals generates broad and potent antibody responses. Here, we isolate 459 spike-specific monoclonal antibodies (mAbs) from two individuals who were infected with the index variant of SARS-CoV-2 and later boosted with mRNA-1273. We characterize mAb genetic features by sequence assignments to the donors' personal immunoglobulin genotypes and assess antibody neutralizing activities against index SARS-CoV-2, Beta, Delta, and Omicron variants. The mAbs used a broad range of immunoglobulin heavy chain (IGH) V genes in the response to all sub-determinants of the spike examined, with similar characteristics observed in both donors. IGH repertoire sequencing and B cell lineage tracing at longitudinal time points reveals extensive evolution of SARS-CoV-2 spike-binding antibodies from acute infection until vaccination five months later. These results demonstrate that highly polyclonal repertoires of affinity-matured memory B cells are efficiently recalled by vaccination, providing a basis for the potent antibody responses observed in convalescent persons following vaccination.

The rapid global spread of SARS-CoV-2 has highlighted the need to understand qualitative aspects of our immune response to emerging and evolving viruses, particularly neutralizing antibody activity and the duration of protective immunity. A wealth of studies has shown that SARS-CoV-2-infected individuals respond with rapid IgG production and neutralizing antibodies that are primarily directed against the receptor-binding domain (RBD) of subdomain 1 (S1) of the virus spike (S). The strength of the early response correlates with disease severity, with persons who experience mild symptoms typically producing lower antibody levels than those who develop moderate or severe disease[1–3]. Serum antibody levels decline gradually once viral replication is controlled and short-lived antibody-producing plasma cells are no longer produced. However, antibody affinity maturation in germinal centers (GCs) continues for several months after the infection. This results in an improved quality of the memory B-cell (MBC) compartment, which can be engaged upon re-exposure to antigen[4–6]. Since COVID-19 vaccines became available, many reports have described properties of the elicited immune response; the best-studied vaccines being the mRNA vaccines from Moderna[7] and Pfizer/BioNtech[8]. While these vaccines offer high levels of protection against severe disease, the antibody response wanes, and frequent boosting is required to prevent or reduce symptomatic disease[9,10].

Waning antibody responses and the emergence of multiple SARS-CoV-2 variants of concern (VOCs) that partially or markedly evade

[1]Department of Microbiology, Tumor and Cell Biology, Karolinska Institutet, Stockholm, Sweden. [2]Adimab LLC, Lebanon, NH 03766, USA. [3]Department of Pediatrics, Dartmouth-Hitchcock Medical Center, Lebanon, NH 03756, USA. [4]Department of Clinical Microbiology, Umeå University, Umeå, Sweden. [5]Invivyd Inc, Waltham, MA 02451, USA. [6]These authors contributed equally: Mark Chernyshev, Mrunal Sakharkar, Ben Murrell, Laura M. Walker, Gunilla B. Karlsson Hedestam. ✉e-mail: lwalker@invivyd.com; gunilla.karlsson.hedestam@ki.se

antibody responses elicited by previous infection or vaccination have impeded the establishment of durable protection against the virus. Highly transmissible VOCs such as Delta, Omicron, and newly emerging Omicron subvariants reinforce that SARS-CoV-2 is a continuously evolving pathogen. Studies have shown that the individuals who were first infected with SARS-CoV-2 and then vaccinated (sometimes referred to as hybrid immunity) develop higher antibody titers and increased neutralization breadth against VOCs compared to those who were only infected or vaccinated[11–16].

While serological studies provide critical information about overall antibody titers and neutralization breadth, qualitative studies of memory B cell (MBC) and plasma cell can greatly help our understanding of how the humoral immune response evolves over time. Here, we applied high-throughput monoclonal antibody (mAb) isolation to retrieve 459 spike-binding mAbs from two individuals who were first SARS-CoV-2 infected and later vaccinated with mRNA-1273 (232 mAbs from donor IML3694 and 227 mAbs from donor IML3695), and we characterized these for their genetic (germline gene usage, clonality, SHM) and functional (subdomain specificity and neutralization) properties. We then combined this with deep *IGH* repertoire sequencing (Rep-seq) and mAb linage tracing at longitudinal timepoints to obtain an improved understanding of the dynamics of the response. Of the 459 spike-binding mAbs, a set of mAbs (*n* = 33) bound both the SARS-CoV-2 and the HCoV-HKU1 spike. Most of the cross-reactive mAbs were found at the acute infection timepoint and likely originated from pre-existing MBCs generated by a prior infection with HKU1 or a related beta-CoV. The HKU1 cross-reactive mAbs displayed significantly higher levels of somatic hypermutation (SHM) at the acute infection timepoint than the SARS-CoV-2 S-specific mAbs isolated from the same timepoint. Furthermore, except for a single sequence the HKU1 cross-reactive lineages could not be traced in IgM repertoires from the acute infection timepoint, unlike the SARS-CoV-2 S-specific lineages, which could be traced in the IgM repertoire, consistent with de novo elicitation of the latter. Lineage tracing in total IgG repertoires from longitudinal timepoints demonstrated that the CoV-2 S-specific mAbs diversified and acquired extensive SHM in the 5 months following the infection, resulting in a highly polyclonal MBC pool that was readily recalled and expanded by the vaccination. These results offer a detailed dissection of the B-cell response to SARS-CoV-2 S at a clonal level and over time, illustrating the dynamics of the response in individuals who were infected during the first wave of the pandemic and mRNA vaccinated 5 months later.

## Results
### Prior infection results in significantly increased antibody binding and neutralizing titers upon vaccination
We analyzed serum IgG titers against SARS-CoV-2 and HCoV-HKU1 S in samples from individuals who had recovered from SARS-CoV-2 infection in 2020, both before and after vaccination, compared to uninfected vaccinated individuals and unvaccinated pre-pandemic controls (Supplementary Data 1). Responses were highest in individuals who were previously infected then vaccinated, with half-maximal effective concentration (EC$_{50}$) enzyme-linked immunosorbent assay (ELISA) titers more than an order of magnitude higher than in the same individuals prior to vaccination (Supplementary Fig. 1A). In contrast, antibody titers against HCoV-HKU1 S were similar in the pre-pandemic samples and the samples from uninfected vaccinated individuals, while infected individuals displayed higher HCoV-HKU1 S titers both before and after vaccination (Supplementary Fig. 1B), consistent with previous work[6]. Donors with prior SARS-CoV-2 infection who were later vaccinated displayed higher serum neutralizing titers against all VOCs tested than individuals who were only vaccinated (Supplementary Fig. 1C), as also shown by others[11–16].

### Isolation and characterization of SARS-CoV-2 spike-specific monoclonal antibodies
To investigate the evolution of the B-cell response in more detail, we collected sequential serum and peripheral blood mononuclear cells (PBMCs) from two SARS-CoV-2 recovered individuals, IML3694 and IML3695, who each received a single mRNA-1273 vaccination -5 months after infection with the index virus. We acquired longitudinal samples from each respective donor as follows: 11- or 14 days post infection (acute timepoint), 32- or 37 days post infection (convalescent timepoint), 160- or 169-days post infection (pre-vax timepoint), and 9- or 7 days post-vaccination (post-vax timepoint) (Fig. 1a and Supplementary Data 1). Assessment of serum neutralizing antibodies showed that the two donors had comparable neutralizing activity against the index virus (index), Beta, Delta, and Omicron BA.1. Beta and Omicron BA.1 were more neutralization-resistant than the index and Delta variants (Fig. 1b), as previously reported[17–22]. A total of 459 spike-specific mAbs were isolated across the acute, pre-vax, and post-vax timepoints, 232 from IML3694 and 227 from IML3695 (Supplementary Data 2). The mAbs were isolated from total antibody-secreting cells (ASCs) at the acute and post-vax timepoints and from spike-binding MBCs at the pre-vax timepoint (Supplementary Fig. 2A). The subdomain specificities of the mAbs were determined by binding to recombinant SARS-CoV-2 subdomain 2 (S2), N-terminal domain (NTD) and receptor-binding domain (RBD) proteins. S2-specific antibodies dominated the response in both donors (Fig. 1c), consistent with previous observations[6,23]. Of the small proportion of SARS-CoV-2 and HKU1 S-cross-reactive mAbs, most were isolated at the acute timepoint from donor IML3694 (Supplementary Fig. 2B). In both donors, most of the cross-reactive mAbs were directed against the S2 domain or an undefined non-RBD epitope (Supplementary Fig. 2C). When examining mAbs isolated at the three different timepoints separately, we observed differences in *IGHV* gene usage with proportionally lower use of *IGHV1* and *IGHV4* family genes at the acute infection time compared to the other timepoints (Supplementary Fig. 3A). This was largely explained by the fact that proportionally more HCoV-HKU1-binding mAbs were isolated from this timepoint, many of which used *IGHV3-30* (*IGHV3-30*18*). We also observed a peak of post-vax *IGHV1-69*-using mAbs, which included several (*n* = 21) clonally diverse *IGHV1-69*-using antibody lineages that were readily expanded by vaccination (Supplementary Fig. 3A). 12 of these were paired with *IGKV3-11* and targeted S2 and thus may be of the same class as the antibodies described in ref. 24. This skewing between mAbs isolated at different timepoints was also apparent at the level of subdomain specificities. mAbs isolated from the acute infection timepoint showed a different distribution of subdomain specificities compared to those isolated from pre- and post-vax timepoints due to the frequency of HCoV-HKU1 cross-reactive lineages at the acute timepoint. Despite binding to the full-length trimeric SARS-CoV-2 S, many of these could not be mapped to a specific subdomain, possibly due to low affinity or the lack of quaternary epitopes on the probes used here (Supplementary Fig. 3B). The subdomain specificities of the mAbs isolated at the different timepoints were similar between the two donors (Supplementary Fig. 3C).

We next assigned germline immunoglobulin heavy chain *V*, *D*, and *J* (*IGHV*, *IGHD,* and *IGHJ*) allele usage to each mAb. To ensure correct assignment, personalized *IG* genotyping was performed using IgDiscover[25] to infer germline *IG* alleles from expressed IgM repertoires from each donor (Supplementary Data 3). Correct allelic assignment is required not only for correct clonotyping of antibodies but also for precise calculations of SHM. The SARS-CoV-2 S-specific mAbs used a broad range of *IGHV* genes, with *IGHV3-9*, *IGHV3-30*, *IGHV3-30-3*, *IGHV3-33*, *IGHV4-31*, *IGHV1-69*, and *IGHV4-59* being the most frequently used. We found that the gene usage was highly similar between the two donors, demonstrating inter-donor consistency in the engagement of the B-cell repertoire against SARS-CoV-2 (Fig. 1d).

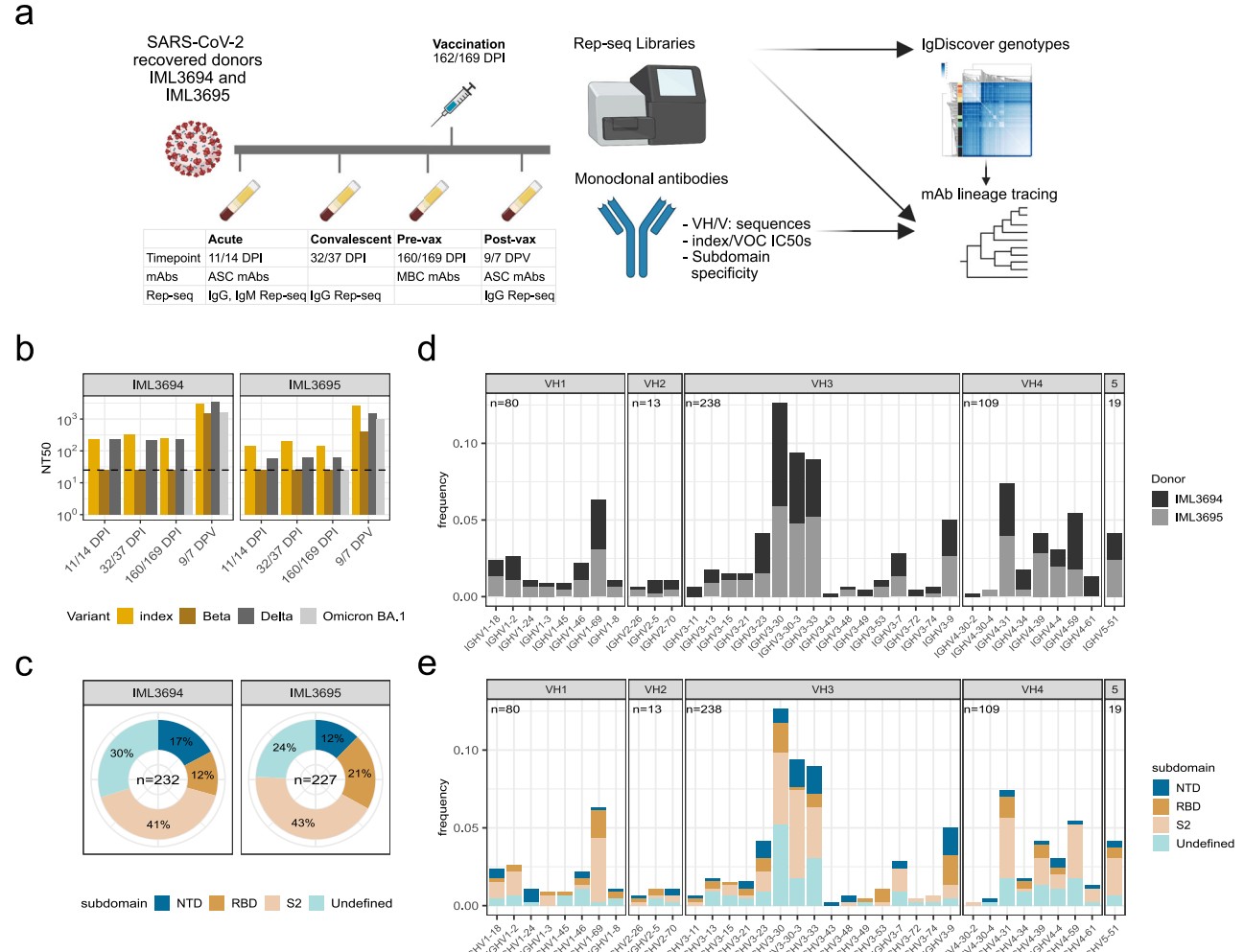

**Fig. 1 | Study design, serum neutralizing activity, and properties of spike-binding monoclonal antibodies (mAbs). a** Schematic of study design with time-points for infection and vaccination, mAb isolation, and samples for Rep-seq. **b** IML3694 and IML3695's serum neutralizing antibody values against index SARS-CoV-2 (index) and variants of concern (VOCs). **c** Pie charts of mAb subdomain specificities in the two donors. **d** mAb *IGHV* allele frequencies colored by donor. **e** mAb *IGHV* allele frequencies colored by subdomain specificity.

Nearly half of the *IGHV* genes were used in response to all three sub-domains, RBD, NTD, and S2. Biased *IGHV* gene usage in the RBD response is consistent with previous work[26–31].

Of the full panel of mAbs described in this study, 31 displayed neutralizing half-maximal inhibitory concentration (IC$_{50}$) values below 0.4 μg/ml against SARS-CoV-2, and four mAbs (ADI-67744, ADI-67138, ADI-67183, and ADI-67857) potently neutralized Omicron BA.1 with IC$_{50}$ values below 0.02 μg/ml (Fig. 2). Since all mAbs were isolated prior to the emergence of the Omicron lineage and prior to the use of Omicron variant vaccines, these results demonstrate the ability of the immune system to generate antibodies against epitopes that are conserved between the index virus spike and VOCs, despite numerous mutations in the RBD of many variants. Of the four Omicron BA.1 neutralizing mAbs, three potently neutralized the Omicron sub-lineage, BA.2.75, yet all failed to neutralize the Omicron sub-lineage BA.5 (Fig. 2).

**Bulk antibody repertoire sequencing for lineage tracing**
To examine the evolution of the B-cell lineages defined by the isolated mAbs, we produced bulk IgG libraries from each donor using PBMCs sampled at the acute infection, convalescent and post-vax timepoints. To analyze B-cell repertoire diversity rather than transcript counts, the Rep-seq *VDJ* nucleotide sequences were deduplicated and denoised using the Fast Amplicon Denoising (FAD) algorithm[32]. An example of a

traced antibody lineage before and after FAD is shown in Supplementary Fig. 4A. We also observed that some of the high SHM sequences were the result of apparent chimeric products, likely formed during PCR, which we filtered out using a chimeric *IGHV* sequence detection method. The effect of chimeric cleaning was visible when examining individual antibody lineages with chimeric sequence average SHM consistently falling above the average SHM for the non-chimeric sequences (Supplementary Fig. 4B). All subsequent tracing analysis was performed following FAD and chimera cleaning. The ratio of clonotypes to unique sequences was higher on average (0.86 with standard deviation of 0.03) for IgM compared to IgG repertoires (0.27 with standard deviation of 0.05), which may be explained by IgG repertoires being more heavily dominated by expanded clones (Supplementary Fig. 4C). We observed that *IGHV3-30* and *IGHV3-33* usage was high across all repertoires, especially for IgG (Supplementary Fig. 4D). *IGHV3-23* was among the most frequently used genes in the IgM repertoires, especially in IML3695, but it was not often used proportionally in the S-specific mAbs.

**Lineage analysis reveals the development of spike-binding antibodies following infection and vaccination**
Of the HCoV-HKU1 cross-reactive mAbs isolated from the different timepoints, 49% were isolated from the acute infection timepoint

| Donor | mAb ID | VH gene | JH gene | HCDR3 | % VH SHM nt | VL gene | JL gene | LCDR3 | % VL SHM nt | index IC50 (µg/ml) | Beta IC50 (µg/ml) | Delta IC50 (µg/ml) | Omicron BA.1 IC50 (µg/ml) | Omicron BA.2.75 IC50 (µg/ml) | Omicron BA.5 IC50 (µg/ml) |
|---|---|---|---|---|---|---|---|---|---|---|---|---|---|---|---|
| IML3694 | ADI-67481 | IGHV3-53*02 | IGHJ4*02 | ARDLDVMGGFDY | 2 | IGKV1-9*01 | IGKJ1*01 | QELNSYPRT | 2.1 | 0.0589 | >1 | 0.1227 | N.D. | N.D. | N.D. |
| IML3694 | ADI-67485 | IGHV1-69*01 | IGHJ2*01 | ATGVRYCDTTSCRASYFDF | 4.7 | IGKV3-20*01 | IGKJ2*01 | QQYGSSPLT | 1.7 | 0.0148 | >1 | 0.2704 | 4.211 | N.D. | N.D. |
| IML3694 | ADI-67649 | IGHV4-4*02 | IGHJ4*02 | ARDIRQDDCSTTRCPEY | 5.1 | IGKV3-11*01 | IGKJ3*01 | QQRSNWPPT | 1.7 | 0.012 | >1 | 0.0302 | N.D. | N.D. | N.D. |
| IML3694 | ADI-67652 | IGHV1-2*02 | IGHJ6*03 | ARDRNWAIFGWESDV | 5.7 | IGLV2-23*02 | IGLJ3*02 | CSYADSSAWV | 4.1 | 0.1698 | >1 | 0.302 | N.D. | N.D. | N.D. |
| IML3694 | ADI-67657 | IGHV3-23*01 | IGHJ4*02 | AKDRLRTSSLQPMPFFDY | 3.7 | IGKV4-1*01 | IGKJ1*01 | QQYYSAPYT | 1.7 | 0.0007 | N.D. | >1 | 2.139 | N.D. | N.D. |
| IML3694 | ADI-67679 | IGHV3-53*02 | IGHJ4*02 | ARDLAGPGLFDH | 5.8 | IGKV1-33*01 | IGKJ5*01 | QQYDNVPSIT | 3.2 | 0.0584 | >1 | 0.0599 | 0.1204 | N.D. | N.D. |
| IML3694 | ADI-67680 | IGHV3-33*01 | IGHJ2*01 | ARDGTIAVAGTFDRFFDL | 3.4 | IGKV1-33*01 | IGKJ3*01 | QQYHILPFT | 3.9 | 0.1108 | >1 | 0.2379 | N.D. | N.D. | N.D. |
| IML3694 | ADI-67722 | IGHV4-34*01 | IGHJ6*03 | ARGVQIPEYCSMNNCPVSDHHYFYMDV | 13.3 | IGKV3-20*01 | IGKJ1*01 | QQYGGSRPWT | 4.5 | 0.0282 | 0.2552 | 0.0122 | N.D. | N.D. | N.D. |
| IML3694 | ADI-67726 | IGHV1-8*01 | IGHJ6*03 | ATKRVQVPRRYYYYMDL | 3.1 | IGKV2-28*01 | IGKJ4*01 | MQSLQTPLT | 2 | 0.0258 | N.D. | 0.0235 | N.D. | N.D. | N.D. |
| IML3694 | ADI-67744 | IGHV3-9*01 | IGHJ6*02 | AKGKAAGHSYYYGMDV | 3.4 | IGKV1-39*01 | IGKJ1*01 | QQSYVTPWT | 3.9 | 0.0154 | 0.0272 | 0.0102 | 0.01366 | 0.007 | >1 |
| IML3694 | ADI-67748 | IGHV4-31*03 | IGHJ4*02 | ARSPVIYGTNSGFDY | 5.7 | IGKV5-2*01 | IGKJ1*01 | LQHDNFPYT | 1.1 | 0.0082 | 0.0322 | 0.0431 | 4.405 | N.D. | N.D. |
| IML3694 | ADI-67757 | IGHV1-8*01 | IGHJ6*02 | ARGNYFDGDGYVDY | 4.1 | IGKV3-15*01 | IGKJ4*01 | QQYNNWPLT | 3.2 | 0.0216 | >1 | 0.0302 | N.D. | N.D. | N.D. |
| IML3694 | ADI-67831 | IGHV3-49*03 | IGHJ4*02 | TLTVTNRYYFHS | 2.4 | IGKV3-15*01 | IGKJ2*03 | QQYNNWFS | 0.7 | 0.009 | >1 | 0.8322 | N.D. | N.D. | N.D. |
| IML3695 | ADI-67119 | IGHV3-53*02 | IGHJ6*03 | ARDSVRGGMDV | 5.8 | IGLV3-9*01 | IGLJ2*01 | QVWDTTSVI | 8.6 | 0.065 | 0.6351 | 0.0808 | N.D. | N.D. | N.D. |
| IML3695 | ADI-67127 | IGHV3-9*01 | IGHJ3*02 | AKDDYPSSWYEHHPQRWAFDI | 3.7 | IGLV1-51*01 | IGLJ3*02 | GTWDSSLSVV | 0.3 | 0.0541 | >1 | >1 | N.D. | N.D. | N.D. |
| IML3695 | ADI-67135 | IGHV3-9*01 | IGHJ4*02 | AKGGEYSSRWYLRESYFDY | 5.1 | IGKV2D-29*01 | IGKJ4*01 | MQSIQVPLT | 1.7 | 0.0025 | N.D. | >1 | N.D. | N.D. | N.D. |
| IML3695 | ADI-67138 | IGHV4-39*01 | IGHJ4*02 | AVGGVRSLEWLLQFDY | 6.1 | IGLV2-8*01 | IGLJ1*01 | SSYAGSSSLV | 2.7 | 0.0065 | 0.001 | 0.003 | 0.002276 | 0.003 | >1 |
| IML3695 | ADI-67139 | IGHV4-31*03 | IGHJ4*02 | ARDSDYGEYFDS | 3.3 | IGKV3-11*01 | IGKJ5*01 | QQRYNWPPIT | 1.7 | 0.2125 | >1 | >1 | N.D. | N.D. | N.D. |
| IML3695 | ADI-67140 | IGHV3-53*02 | IGHJ3*02 | AREAYAFDI | 5.5 | IGKV1-9*01 | IGKJ4*01 | QQLNSHPPA | 2.1 | 0.012 | >1 | 0.015 | N.D. | N.D. | N.D. |
| IML3695 | ADI-67183 | IGHV4-39*01 | IGHJ4*02 | AVGGVRFFEWLLQFDY | 3.7 | IGLV2-8*01 | IGLJ2*01 | SSYAGSSSLI | 3.7 | 0.0029 | 0.0027 | 0.0008 | 0.003034 | 0.005 | >1 |
| IML3695 | ADI-67189 | IGHV3-30*03 | IGHJ4*02 | ARDYGDYAAFDS | 4.4 | IGKV4-1*01 | IGKJ4*01 | QQYYSTPLT | 1.3 | 0.2776 | >1 | >1 | N.D. | N.D. | N.D. |
| IML3695 | ADI-67218 | IGHV4-34*01 | IGHJ3*02 | ARWDLLYPRDAFDI | 3.1 | IGLV2-11*01 | IGLJ2*01 | CSYAGSYVV | 2.1 | 0.0657 | >1 | 0.057 | N.D. | N.D. | N.D. |
| IML3695 | ADI-67852 | IGHV3-9*01 | IGHJ4*02 | AKDIQFRDRDCTNGVCSVGGFDY | 5.7 | IGKV3-15*01 | IGKJ2*01 | QQYNKWPPRT | 1.7 | 0.2345 | >1 | >1 | 1.103 | N.D. | N.D. |
| IML3695 | ADI-67857 | IGHV4-39*01 | IGHJ4*02 | AVGGVRFLEWLLQFDY | 2.4 | IGLV2-8*01 | IGLJ1*01 | SSYAGSSNLV | 1.7 | 0.0027 | 0.0727 | 0.0159 | 0.01917 | 0.089 | >1 |
| IML3695 | ADI-67935 | IGHV4-31*03 | IGHJ4*02 | ARGPYASGSFDS | 5 | IGKV3-20*01 | IGKJ1*01 | HHYGSSGDT | 2.1 | 0.048 | N.D. | 0.039 | N.D. | N.D. | N.D. |
| IML3695 | ADI-67971 | IGHV3-9*01 | IGHJ3*02 | AKLGGAN.D.YDFRSGPTAFDI | 4.4 | IGLV1-44*01 | IGLJ2*01 | ATWDDSLNGVVV | 2 | 0.1363 | 0.3985 | >1 | 0.1333 | N.D. | N.D. |
| IML3695 | ADI-67977 | IGHV3-9*01 | IGHJ6*02 | VKDMSVGDRSVEYYGMDV | 2 | IGLV3-21*04 | IGLJ2*01 | QVWDSSSENVV | 2.1 | 0.0759 | 0.2904 | 0.4427 | 0.02833 | N.D. | N.D. |
| IML3695 | ADI-67981 | IGHV1-69*01 | IGHJ3*02 | ARDGRHNYDSTGYYHN.D.FDI | 5.7 | IGKV1-39*01 | IGKJ1*01 | QQSYSSRT | 3.2 | 0.0094 | >1 | 0.0117 | 0.04438 | N.D. | N.D. |
| IML3695 | ADI-67983 | IGHV3-9*01 | IGHJ4*02 | AKGKWPSSPSFLTDY | 5 | IGKV1-27*01 | IGKJ4*01 | QKYNSVPLT | 3.5 | 0.0031 | >1 | 0.0049 | N.D. | N.D. | N.D. |
| IML3695 | ADI-67994 | IGHV3-23*01 | IGHJ6*02 | ARDLGGYSYGLNFFYAMDV | 3.8 | IGLV2-14*03 | IGLJ2*01 | SSYTSSSTFVL | 2 | 0.1967 | 0.3379 | 0.227 | 0.3084 | N.D. | N.D. |
| IML3695 | ADI-67999 | IGHV1-69*01 | IGHJ4*02 | ARERGYSGYGASLYFDY | 5.1 | IGLV1-40*01 | IGLJ1*01 | QSYDSSLSGAV | 1.7 | 0.5019 | 0.6414 | >1 | 0.1381 | N.D. | N.D. |

N.D.: Not Done

**Fig. 2 | Properties of SARS-CoV-2 neutralizing mAbs.** An $IC_{50}$ threshold of 0.4 µg/ml against any of the SARS-CoV-2 variants qualified mAbs as neutralizing. The table includes heavy and light chain VDJ assignments as well as SHM values and $IC_{50}$s against 6 SARS-CoV-2 variants.

(Fig. 3a). When comparing SHM levels for the HCoV-HKU1 cross-reactive mAbs ($n = 16$) with SARS-CoV-2 spike-specific mAbs ($n = 50$) from the acute infection timepoint, we observed a significant difference in median SHM levels: 7.56% compared to 3.49%, respectively, at the nucleotide level (Fig. 3b). This suggested that pre-existing HCoV-HKU1 memory B cells were reactivated by SARS-CoV-2 infection in these donors, consistent with the serological data (Supplementary Fig. 1) and previous reports[33]. To investigate this further, we selected an HCoV-HKU1 spike cross-reactive lineage, ADI-66175, for tracing in the IgG Rep-seq data. The ADI-66175 lineage was represented by 7 clonally related mAbs from the complete mAb set (Supplementary Data 2).

Tracing the expanded HCoV-HKU1 S-cross-reactive lineage, ADI-66175, in the IgG Rep-seq data revealed that it evolved extensively during the acute SARS-CoV-2 infection. This lineage was especially rich with numerous somatic variants traced at the acute infection timepoint and a smaller number at the convalescent timepoint (Fig. 3c). The full range of SHM detected for this lineage suggested that a range of variants, from unmutated to intermediate to highly mutated, were archived in memory B cells. We did not identify any members of this lineage in the IgM libraries, supporting the conclusion that this was a pre-existing IgG-switched lineage.

We next evaluated three SARS-CoV-2 S-specific lineages for which large numbers of somatic variants were traced, ADI-66196 and ADI-67860, S2-specific Abs, and ADI-67983, an RBD-specific neutralizing Ab. ADI-66196 was isolated at the acute infection timepoint and had very low SHM, while ADI-67860 and ADI-67983 were isolated from the pre-vax timepoint and had higher SHM. Variants of the three mAbs were traced in both IgM and IgG libraries from the acute infection timepoint when both ASCs and MBCs are likely to be present, while almost no sequences could be traced at the convalescent timepoint, suggesting that the memory B-cell pool was small or contracted. However, extensive expansion was observed for both lineages at the post-vax timepoint with multiple distinct evolutionary branches as demonstrated by phylogenetic tree analysis (Fig. 4a). SHM comparisons between the mAbs and the traced sequences illustrated the increase in SHM between the acute infection timepoint and the post-vax timepoint (Fig. 4b). Because the post-vax timepoint was as early as 7 or 9 days after vaccination, the high SHM levels observed at this time likely reflected affinity maturation that had occurred in the time leading up to the vaccination, archived in memory B cells, rather than an increase in SHM following vaccination. The fact that ADI-67983, which was isolated at the pre-vax timepoint, had similar SHM levels as its traced somatic variants at the post-vax timepoint illustrates this point. Furthermore, these analyses highlight that inclusion of IgM in lineage tracing bridged the gap between disparate clades of IgG sequences, aiding the construction of lineage trees.

## Vaccination expands a broad range of infection-induced S-specific Ab lineages
Having confirmed that SARS-CoV-2-specific B-cell lineages could be traced in bulk Rep-seq data from the post-vax timepoints for selected mAb sequences, we performed a broader search that included all infection-induced SARS-CoV-2 S-specific Ab lineages, defined either by mAbs that were isolated prior to vaccination, or by mAbs that were isolated from the post-vax timepoint, but which could be traced back to a timepoint prior to the vaccination. Clonal collapsing revealed that the 459 mAbs belonged to 405 mAb lineages of which 200 (49.4%) were traceable in the Rep-seq data at one or several timepoints, including 13 neutralizing lineages. Traced lineages were assigned to the subdomain specificities of the corresponding mAbs, which resulted in a total of 82 S2 lineages, 33 NTD lineages, 32 RBD lineages, and 53 lineages with undefined subdomain specificity. The subdomain specificity proportions among lineages were consistent with the proportions among the mAbs, with S2 lineages being the most common and NTD and RBD present at similar numbers.

Strikingly, analysis of the IgG libraries generated from day 9 (IML3694) and day 7 (IML3695) after vaccination demonstrated that vaccination expanded a broad range of infection-induced B-cell lineages, with 88 clonally distinct lineages in the two donors. This included lineages defined by mAbs isolated from MBCs at the pre-vax timepoint ($n = 52$) and lineages defined by mAbs that were isolated at other timepoints and traced both before and after the vaccination

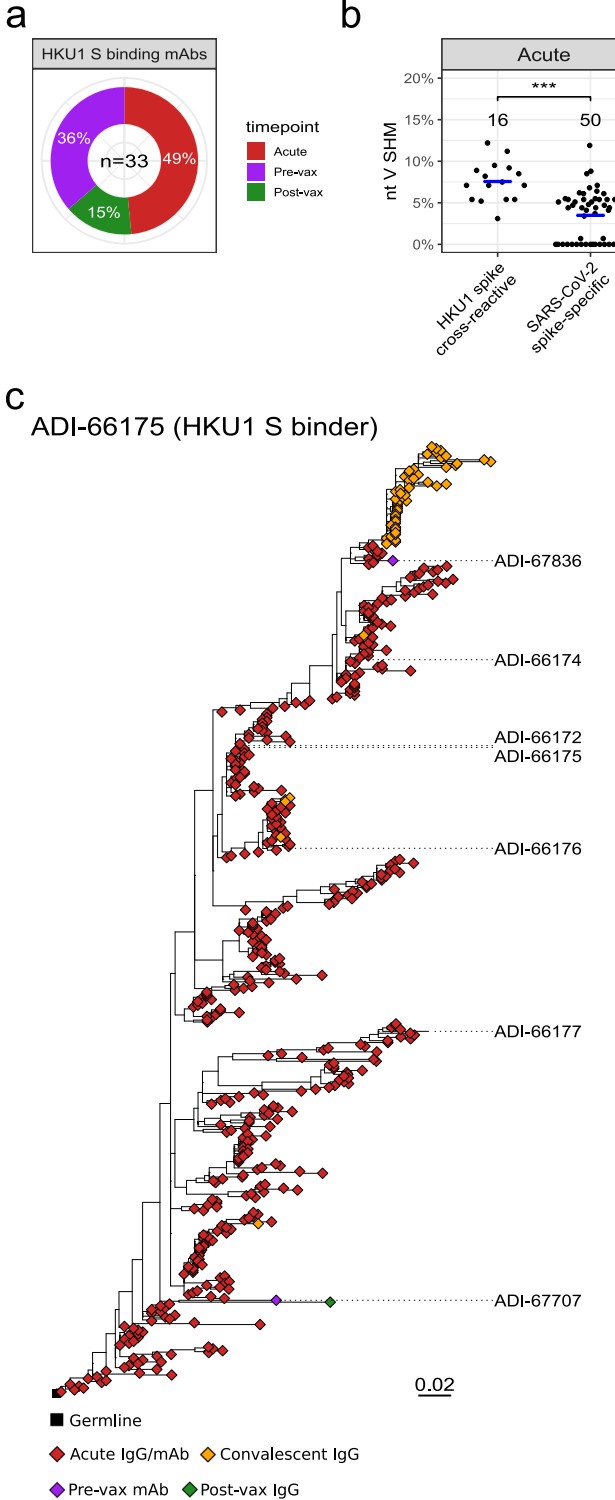

**a**

HKU1 S binding mAbs

n=33

49% Acute
36% Pre-vax
15% Post-vax

timepoint
Acute
Pre-vax
Post-vax

**b**

Acute

16  50

HKU1 spike cross-reactive / SARS-CoV-2 spike-specific

nt V SHM

***

**c**

ADI-66175 (HKU1 S binder)

ADI-67836
ADI-66174
ADI-66172
ADI-66175
ADI-66176
ADI-66177
ADI-67707

0.02

■ Germline
◆ Acute IgG/mAb
◆ Convalescent IgG
◆ Pre-vax mAb
◆ Post-vax IgG

**Fig. 3 | Isolation timepoints and evolution of HCoV-HKU1 S-cross-reactive mAbs. a** Pie chart of timepoints at which HCoV-HKU1 S-binding mAbs were isolated. **b** Dot plot of % nucleotide *IGHV* SHM for the HCoV-HKU1 S-cross-reactive and SARS-CoV-2 S-specific mAbs isolated at the acute timepoint. A two-sided Mann−Whitney *U* test was used for comparison; *** indicates a *P* value < 0.001, with the exact value being *P* value = 1.261e-5. **c** Maximum-likelihood phylogenetic trees of HCoV-HKU1 S-binding traced lineage ADI-66175. The germline sequence was obtained from the IgBLAST-generated "germline_alignment" column of the sequence with the minimum *IGHV* SHM in the lineage.

($n$ = 36). Overall, these results suggest that the vaccine-induced recall response engaged a highly polyclonal B-cell repertoire, including several neutralizing Ab lineages (Fig. 5a). For most lineages, we observed a decrease in traceable sequences during the convalescent timepoint compared to the acute timepoint, which may be explained by either a contraction in SARS-CoV-2 spike-specific MBCs or a lack of circulating S-specific ASCs at this timepoint. Interestingly, we identified a set of S-specific lineages (ADI-66210, ADI-66213, ADI-66196, ADI-66197, ADI-67108, ADI-67109, and ADI-67112) that were present both at the acute infection timepoint, when these mAbs were isolated from ASCs, and as MBCs that could be reactivated by the vaccination 5 months later, suggesting dual differentiation fates for infection-induced B cells.

When we examined the median level of SHM in lineages that were traced in both IgM Rep-seq data from the acute infection timepoint and IgG Rep-seq data from the post-vax timepoint there was a clear increase in SHM in all lineages (Fig. 5b, left). A similar effect was observed in lineages traced in IgG Rep-seq data from the acute infection timepoint and IgG Rep-seq data from the post-vax timepoint (Fig. 5b, right). We also compared SHM levels of the mAbs that were isolated from MBCs at the pre-vax timepoint with the median SHM of their variants that were traced at the post-vax timepoint. We found the SHM levels to be highly similar, consistent with affinity maturation having occurred during the time leading up to the vaccination rather than in the short period (9 or 7 days) between the vaccination and the post-vax sampling timepoint (Fig. 5c). Thus, the peak in serum antibody titers observed 1 week after vaccine boosting (Fig. 1b) reflects antibody sequences archived in MBCs and their differentiation into antibody-secreting plasmablasts upon antigen restimulation.

## Discussion

The SARS-CoV-2 pandemic is unique due to its rapid spread and extensive impact on societies, and because vaccines were developed and deployed within just a year of isolating the virus. Vaccination has curtailed the damage caused by the pandemic by protecting individuals from severe disease. Two mRNA-based vaccines, mRNA-1273 from Moderna and BNT162b2 from Pfizer/BioNTech, were among the first COVID-19 vaccines to be approved, with several inactivated virus- and vector-based vaccines gaining approval around the same time or soon thereafter. Thus far, several billion doses of mRNA vaccines have been administered globally. The availability of vaccines was initially limited and, while this has improved, access to vaccine doses remains constrained in many parts of the world. Thus, with the continued spread of the virus, the first exposure to SARS-CoV-2 for many persons is natural infection, making studies of how convalescent persons respond to subsequent vaccination highly relevant. Several studies have shown that immunity induced by infection followed by vaccination stimulates higher antibody titers than either alone[11,14,15], but how the B-cell response evolves following infection and vaccination remains a question of interest.

To investigate this question, we used a high-throughput mAb isolation platform to generate 459 spike-specific mAbs representing 405 clonally unique Ab lineages from two individuals who were first infected with index SARS-CoV-2 and subsequently vaccinated with a single dose of mRNA-1273 about 5 months later. Characterization of mAbs isolated from different timepoints following infection or vaccination revealed that they used a broad set of *IGHV* genes to target all subdomains of the spike analyzed (RBD, NTD, and S2), illustrating the polyclonal nature of the response. We identified 31 mAbs that neutralized one or more SARS-CoV-2 variants, including four mAbs that potently neutralized the Omicron BA.2.75 variant. Neutralization of BA.5, which was responsible for the fifth wave of infections in South

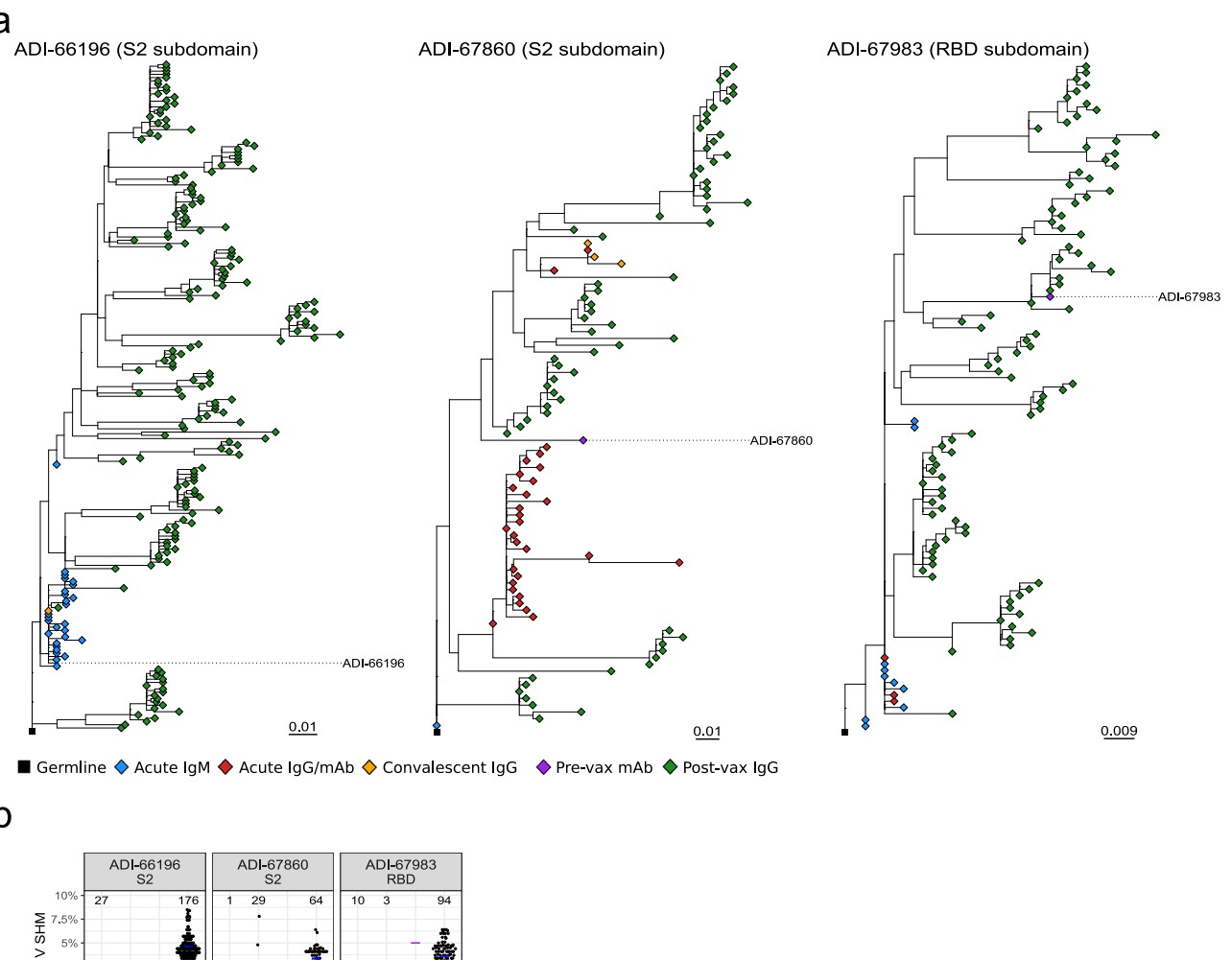

**Fig. 4 | Evolution of SARS-CoV-2 S-specific antibody lineages over the sampling timepoints. a** Maximum-likelihood phylogenetic tree of traced lineages containing IgM Rep-seq sequences: ADI-66196, ADI-67860, and ADI-67983. Germlines sequences were obtained from the IgBLAST-generated "germline_alignment" column of the sequence with the smallest *IGHV* SHM in the lineage. **b** Dot plot of % nucleotide *IGHV* SHM for the ADI-66196, ADI-67860, and ADI-67983 lineage sequences. Purple crossbars represent mAb values, while the blue crossbars represent average values of traced variants for each timepoint.

Africa[34] was weak, if detectable. BA.5 escape from neutralizing antibodies induced by index SARS-CoV-2 is well-documented[35,36] and our results are consistent with recent findings showing that BA.2.75 is, for some antibodies, more neutralization-sensitive than BA.5[21]. All neutralizing mAbs isolated here targeted the RBD, the main neutralizing determinant of SARS-CoV-2, although neutralizing mAbs that target the NTD have been demonstrated in other studies[37].

A set of non-neutralizing mAbs (*n* = 33) were cross-reactive with the spike of HCoV-HKU1, the majority of which were isolated at the acute infection timepoint when total ASCs were used as the source for mAb isolation. Consistent with prior studies, the cross-reactive mAbs displayed overall higher levels of SHM than the SARS-CoV-2 spike-specific mAbs isolated at the same timepoint, suggesting that SARS-CoV-2 infection boosted pre-existing cross-reactive MBCs induced by prior infection with endemic HCoVs, such as HKU1[6,33,38]. The HCoV-HKU1 cross-reactive Ab lineage that we studied in-depth, ADI-66175, evolved additional SHM through the convalescent phase, but was not boosted by vaccination, suggesting that it recognized an epitope that was not present on the vaccine antigen. This is consistent with results reported elsewhere showing that mRNA vaccination did not robustly

boost pre-existing antibodies to endemic HCoVs[39]. Other studies have shown that HCoV cross-reactive Abs are mostly S2-directed, and such Abs are, with rare exceptions[40], non-neutralizing. Whether HCoV cross-reactive non-neutralizing antibody responses play a role in protection against SARS-CoV-2 remains unknown. In our study, 32 of the 33 HCoV-HKU1 cross-reactive mAbs bound non-RBD epitopes and were non-neutralizing. This does not exclude the possibility that they could contribute to protection via antibody-dependent cellular cytotoxicity (ADCC) or other Fc-dependent immune functions that were not studied here.

We found that many SARS-CoV-2 infection-induced Ab lineages that were expanded by the vaccination were S2-directed. Antibody immunodominance was previously shown to shift away from S2 and toward the RBD following Omicron breakthrough infection in vaccinated individuals[41]. The authors attributed this to serum epitope masking by pre-existing S2-specific antibodies relative to the more divergent BA.1 RBD. Thus, it is likely that the type (homologous vs. heterologous exposure) and/or interval between exposures influences the B-cell immunodominance hierarchy at the individual level.

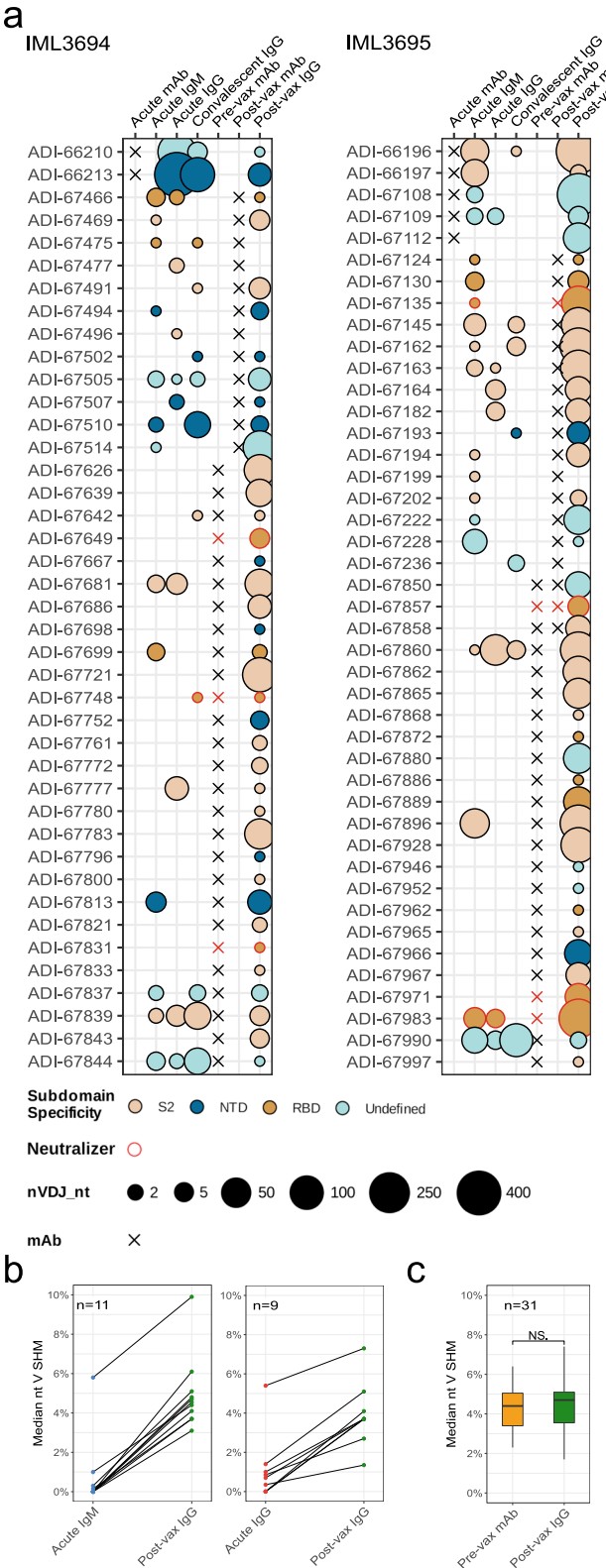

**Fig. 5 | IgM and IgG repertoire sequencing and lineage tracing of SARS-CoV-2 S-specific antibody lineages at the longitudinal timepoints. a** Bubble plots of SARS-CoV-2 S-specific mAb lineages found before and after vaccination in IML3694 and IML3695. The bubble sizes correspond to post-FAD and chimera-cleaned data. **b** Line plots of median % nucleotide *IGHV* SHM for lineages found in acute IgM/IgG and post-vax IgG Rep-seq data. **c** Boxplot of median % nucleotide *IGHV* SHM for lineages containing pre-vax timepoint mAbs and post-vax IgG Rep-seq. The box represents the median ± interquartile range and the whiskers represent the maximum and minimum values. A two-sided Wilcoxon signed-rank test was used for comparison; *P* value = 0.3107.

deep-sequenced IgG repertoires from the acute infection timepoint, as well as from the pre- and post-vax timepoints. The post-vax timepoint, ~1 week after vaccination, was selected to capture the burst of short-lived ASCs that originate from pre-existing MBCs[42,43]. The fact that we could trace as many as 88 SARS-CoV-2 S-specific infection-induced antibody lineages at the post-vax timepoint demonstrates that mRNA-1273 vaccination of previously SARS-CoV-2-infected individuals stimulates a very broad repertoire of MBCs generated from the infection. In both donors, this included RBD-targeting neutralizing Ab lineages, ADI-67649, ADI-67748, and ADI-67831 in IML3694 and ADI-67135, ADI-67857, ADI-67971, and ADI-67983 in IML3695. Additional but less expanded RBD-neutralizing lineages were also observed post-vax. The lineage tracing allowed us to identify many clonal variants of each antibody lineage, and construction of phylogenetic trees illustrated the high degree of diversification and maturation that each lineage undergoes during the months following SARS-CoV-2 infection. Similar studies have been performed in influenza-vaccinated subjects, also demonstrating continued evolution of pre-existing influenza-specific B-cell lineages following vaccination[44].

While it is not expected that all lineages that are present in a donor can be traced from a single blood draw due to sampling limitations, we conclude that many lineages archived in the MBC pool could be measured at the post-vax timepoint when they likely contributed to the increase serum antibody titers. Several mAbs isolated from MBCs sampled at the pre-vax timepoint that were traced at the post-vax timepoint, which was 9 and 7 days later for IML3694 and IML3695, respectively, had similar SHM levels between these two timepoints. These results are consistent with that sequences traced at the post-vax timepoint originate from reactivated MBCs that differentiate into plasmablasts 7 to 9 days after the vaccination without having undergone further affinity maturation in the GC. Whether infection-induced spike-specific IgG-switched MBCs are recruited back to GCs upon vaccine boosting was not investigated here due to the lack of later sampling timepoints for these donors. While it has been shown that increases in SHM within given B-cell lineages result from persistent GCs over long periods of time[45], the extent to which circulating MBCs are recruited back into GCs in response to antigen boosting remains a topic of intense research[46]. Furthermore, in addition to the circulating blood compartment sampled here, MBCs also reside in tissues[47,48]. As such compartments were not sampled here, this study likely underestimates the diversity of SARS-CoV-2 S-specific Ab sequences that were recalled by the vaccination.

Overall, our approach involving mAb isolation coupled with lineage tracing in bulk IgG libraries allowed us to comprehensively probe the SARS-CoV-2 S-specific MBC repertoire. The results underscore the polyclonal nature of the infection-induced Ab response and show that vaccination of convalescent persons expands a broad repertoire of Ab lineages, underpinning the effective responses observed in hybrid immunity.

## Methods
### Study design and donor information
The results shown in Supplementary Fig. 1A were performed under permit 2021-00055 and amendment 2021-01387 approved by the

The greatest benefit of this collection of mAbs was the opportunity to use their sequences to interrogate the B-cell repertoire longitudinally. A primary objective of the study was to determine if there were constraints in the recall response that skewed or limited the post-vax response to a more oligoclonal repertoire. Boosting a broad repertoire of B cells is desired as this provides improved chances to maintain effective neutralizing antibody responses in the face of viral evolution and the emergence of VOCs. To address this question, we

Swedish ethics review authority and is part of an ongoing observational clinical trial to investigate immune responses to Covid-19 vaccination (EudraCT number 2021-000683-30). Inclusion of individuals into our prospective open cohort was through written informed individual consent. At inclusion, we excluded individuals with medication or comorbidities that have an obvious effect on the immune system (i.e., immunosuppressive medication or pid's). Hence, this minimizes the potential of self-selection bias that can affect the results in the study. For the unexposed-vaccinated group, we excluded individuals with self-perceived (and/or diagnosed) SARS-CoV-2 infection. The study participants did not receive any compensation and sex was self-reported. Two other SARS-CoV-2-infected patients were recruited for in-depth longitudinal studies, both of whom gave written informed consent in accordance with the Dartmouth–Hitchcock Hospital (D-HH) Human Research Protection Program (Institutional Review Board) and approved by the Swedish ethics review authority, permit 2021-01850. SARS-CoV-2 infection was confirmed by reverse transcriptase polymerase chain reaction (RT-PCR) after nasal swab in October 2020. Participants received the first dose of the mRNA-1273 vaccine (Moderna) approximately 5 months after the first positive test. Blood samples were collected and fractionated by the Clinical Research Unit of D-HH to obtain PBMCs and serum. The two SARS-CoV-2-infected patients were recruited due to their manageability of symptoms, not requiring hospitalization and willingness to provide blood samples at requested timepoints. The age of both donors is on the higher end of typical "adult" age classification, thus increasing the possibility of prior exposures to circulating seasonal endemic coronaviruses. Donor information and sample collection dates are shown in Supplementary Data 1.

## Recombinant antigens

Prefusion-stabilized SARS-CoV-2 spike protein (S-2P) was generated using a plasmid encoding residues 1–1208 of the SARS-CoV-2 spike. Other features of the plasmid include a mutated S1/S2 furin cleavage site (RSAR to GSAS), proline substitutions at positions 986 and 987, a C-terminal T4 fibritin domain, HRV3C cleavage site, 8x HisTag and TwinStrepTag. HEK-293 cells (DSMZ, ACC 305) were transfected using PEIpro (PolyPlus, Cat# 115-100), followed by the addition of Kifunensine (5 µM) after 3-h. Cell supernatants were harvested, and expressed protein was purified using NiNTD Sepharose resin (Cytiva, Cat# 17531804) and StrepTactin XT Superflow high-capacity resin (IBA Life Sciences, Cat# 24030025). Using size-exclusion chromatography, purified protein was polished successively on a HiLoad 16/600 Superdex 200 pg column (Cytiva, Cat#28989335) and HiLoad 16/600 column packed with 125 mL of Superose 6 resin (Cytiva, Cat# 17048901). Plasmids encoding residues 319-591 of the SARS-CoV-2 spike with a C-terminal HRV3C cleavage site, monomeric Fc-tag and 8x HisTag (SARS-CoV-2 RBD-SD1); residues 1-305 of the SARS-CoV-2 spike with a C-terminal HRV3C cleavage site, monomeric Fc-tag and 8x HisTag (SARS-CoV-2 NTD) were transfected into FreeStyle293F cells (ThermoFisher, R79007) using polyethylenimine. Cell supernatants were harvested after 6 days and purified using Protein A resin (Pierce). Affinity-purified SARS-CoV-2 RBD-SD1 and NTD proteins were then further polished by size-exclusion chromatography on a Superdex 200 Increase column (Cytiva) in a buffer composed of 2 mM Tris pH 8.0, 200 mM NaCl and 0.02% NaN3. The SARS-CoV-2 spike S2 protein was purchased from Acro Biosystems (Cat# S2N-C52H5), non-stabilized SARS-CoV-2 S (Cat# 40589-V08B1) and HCoV-HKU1 S (Cat# 40606-V08B) protein were purchased from Sino Biological.

## Detection and sorting of single B cells

Acute samples post infection and samples post-vaccination were stained to isolate antibody-secreting cells (ASCs), whereas pre-vaccination PBMCs were stained for memory B cells (MBCs). For ASC sorts, PBMCs were stained using anti-human CD19 (PE-Cy7; Biolegend, Cat#

302216) and CD20 (APC-Cy7; Biolegend, Cat#302313), each at 1:1000 dilution; CD38 (PE; Biolegend, Cat# 303506) at 1:400 dilution and CD3 (PerCP-Cy5.5; Biolegend, Cat# 300430), CD8 (1:100; PerCP-Cy5.5; Biolegend, Cat# 344710), CD14 (PerCP-Cy5.5; Invitrogen, Cat# 45-0149-42), CD16 (PerCP-Cy5.5; Biolegend, Cat# 360712), IgM (BV711; BD Biosciences, Cat# 747877), CD71 (APC; Biolegend, Cat# 334107), CD27 (BV510; BD Biosciences Cat# 740167), and propidium iodide (PI) each at 1:100 dilution each. For MBC sorts, PBMCs were stained with CD19 (PE-Cy7; Biolegend, Cat# 302216) at 1:1000 dilution and CD3 (PerCP-Cy5.5; Biolegend, Cat# 300430), CD8 (PerCP-Cy5.5; Biolegend, Cat# 344710), CD14 (PerCP-Cy5.5; Invitrogen, Cat# 45-0149-42), CD16 (PerCP-Cy5.5; Biolegend, Cat# 360712), IgM (BV711; BD Biosciences, Cat# 747877), CD71 (APC-Cy7; Biolegend, Cat# 334110), CD27 (BV510; BD Biosciences, Cat# 740167), PI at 1:100 dilution each and a freshly prepared mixture of PE- and APC-labeled SARS-CoV-2 S-2P protein tetramers (25 nM each). ASCs, defined as $CD19^+CD20^{lo}CD38^+CD27^+CD3^-CD8^-CD14^-CD16^-PI^-$ or class-switched B cells, defined as $CD19^+CD3^-CD8^-CD14^-CD16^-PI^-IgM^-IgD^-$ cells that showed reactivity to both SARS-CoV-2 S-2P tetramers, were single-cell index sorted using a BD FACS Aria II Fusion (BD Biosciences) into 96-well polypropylene microplates (Corning Cat# 07-200-95) containing 20 µl/well of lysis buffer [5 µl of 5× first strand SSIV cDNA buffer (Invitrogen Cat # 18090050B), 0.25 µl RNaseOUT (Invitrogen Cat#10777019), 0.625 µl of NP-40 (Thermo Scientific Cat# 85124), 1.25 µl dithiothreitol (Invitrogen), and 12.85 µl dH2O]. Plates were spun down at 1000×g for 30 s and stored at −80 °C until use. Flow cytometry data were analyzed using FlowJo software v10.8.1.

## Amplification and cloning of antibody variable genes

Human antibody variable gene transcripts (VH, Vκ, Vλ) were amplified by RT-PCR using SuperScript IV enzyme (Thermo Scientific Cat# 18090050) followed by nested PCR using HotStarTaq Plus DNA Polymerase (Qiagen Cat# 203646) and a mixture of IgM-, IgD-, IgA-, and IgG-specific constant-region primers as previously described by Wec et al. Science 2020 and included in Supplementary Data 2. The primers used in the second round of nested PCR contained 40 base pairs of 5′ and 3′ homology with linearized yeast expression vectors to allow cloning by homologous recombination. Amplified transcripts were transformed into S. cerevisiae using the lithium acetate method for chemical transformation[49]. Per transformation reaction, yeast cells (1 × 10[7]) were incubated with a mixture of 240 µl of polyethylene glycol (PEG) 3350 (50% w/v) (Sigma-Aldrich, Cat# 202444), 36 µl of 1 M lithium acetate (Sigma-Aldrich, Cat# 517992), 10 µl of denatured salmon sperm DNA (Invitrogen, Cat# 15632011), 67 µl sterile water, 200 ng of each of the digested vectors and 10 µl each of unpurified VH and VL amplified PCR product at 42 °C for 45 min. Yeast were then washed twice with sterile water, recovered in selective media, and plated for Sanger sequencing.

## Expression and purification of IgG and Fab proteins

To produce monoclonal antibodies (mAbs) as full-length IgG1 proteins, S. cerevisiae yeast cultures were incubated in 24-well plates at 30 °C and 80% relative humidity with shaking at 650 RPM in Infors Multitron shakers. Culture supernatants were harvested after 6 days and IgGs were purified by protein A-based affinity chromatography followed by elution using 200 mM acetic acid with 50 mM NaCl (pH 3.5) and finally neutralized with 1/8 (v/v) 2 M HEPES (pH 8.0). To generate Fab fragments, IgGs were digested with papain at 30 °C for 2-h and the reaction terminated using iodoacetamide. To remove Fc fragments and undigested IgG, the mixtures were passed over protein A agarose. The flow-through was then passed over CaptureSelect™ IgG-CH1 affinity resin (ThermoFisher Scientific) and the Fabs captured on the resin surface were eluted using 200 mM acetic acid with 50 mM NaCl (pH 3.5) followed by neutralized 1/8 (v/v) 2 M HEPES (pH 8.0).

## Biolayer interferometry kinetic measurements

Apparent equilibrium dissociation constant ($K_D^{App}$) affinities were calculated by BLI using a ForteBio Octet HTX instrument (Molecular Devices) as previously described[50]. Reagents were formulated in PBSF (PBS with 0.1% w/v BSA), and all binding steps were performed at 25 °C with 1000 rpm orbital shaking speed. To measure IgG binding to recombinant antigens, IgGs (100 nM) were captured on anti-human IgG (AHC) biosensors (Molecular Devices) and then equilibrated in PBSF for a minimum of 30 min. Following a 60-s baseline step in PBSF, the IgG-loaded biosensors were exposed to the antigen at 100 nM for 180-s and then dipped into PBSF to measure any dissociation of the antigen from the biosensor surface over a period of 180-s. For binding responses >0.1 nm, data were aligned, inter-step corrected (to the association step), and fit to a 1:1 binding model using the ForteBio Data Analysis Software, version 11.1. Only mAbs that bound SARS-CoV-2 S were included in the study.

## SARS-CoV-2 spike-pseudotyped MLV neutralization assay

Single-cycle infection pseudoviruses were generated as previously described[51]. Briefly, HEK-293 cells (DSMZ, ACC 305) were co-transfected with 0.5 μg of the index SARS-CoV-2 spike (NC_045512; pCDNA3.3) or SARS-CoV-2 variants (Beta, Delta, Omicron BA.1) and 2 μg each of MLV luciferase (Vector Builder) and MLV gag/pol (Vector Builder) plasmids using Lipofectamine 2000 (ThermoFisher Scientific). Cell supernatants were harvested 48-h post-transfection and aliquoted to be frozen at −80 °C. To measure neutralizing activity of mAbs, 10,000-15,000 HeLa-hACE2 cells/well (BPS Bioscience Cat #79958) were seeded overnight in 96-well tissue culture plates (Corning). Serial dilutions of mAbs (4 mg/ml-0.5 ng/ml) in cell culture media were incubated with a fixed volume of MLV particles for 1-h at 37 °C, 5% $CO_2$. After washing HeLa-hACE2 cells three times with DPBS, the virus-mAb mixture was directly added over the cells and incubated. After 72-h, the supernatant was aspirated and cells were lysed with Luciferase Cell Culture Lysis 5× reagent (Promega, Cat# E153A). Luciferase activity was measured using the Luciferase Assay System (Promega, Cat# E151A) following the manufacturer's instructions, and relative luminescence units (RLU) were quantified on a luminometer (Perkin Elmer). Percent neutralization was calculated as $100*(1–RLU_{sample}/RLU_{isotype\ control\ mAb})$, and the 50% neutralization concentration was interpolated using four-parameter nonlinear regression fitted curves in GraphPad Prism.

## VSV-SARS-CoV-2 pseudovirus neutralization assay

Microneutralization assays were also performed using a VSV-based pseudovirus system as previously described[52] *Front. Immun.,* 2020). Briefly, mAbs or serum were diluted in twofold series and incubated with VSV-SARS-CoV-2 pseudoviruses for 1 h at 37 °C. The mixture was then added to 293T-hsACE2 cells (Integral Molecular, Philadelphia, PA, C-HA102) and incubated at 37 °C, 5% $CO_2$ for 24 h. Cell lysates were collected and luciferase activity measured using the Bright-Glo system (Promega) with a Bio-Tek II plate reader. Percent neutralization was calculated as [100 − (*mean RLU test wells/mean RLU positive control wells*) × 100] and further used to determine the half-maximal inhibitory concentrations for mAbs (IC$_{50}$) and 50% neutralization titers for serum (NT$_{50}$).

## Enzyme-linked immunosorbent assays (ELISA)

For serum-binding studies, SARS-CoV-2 S-2P protein was diluted to 5 μg/ml in PBS (pH 7.4) and used to coat 96-well high-binding polystyrene ELISA plates (Corning, Cat# 3690) were coated with 25 μl per well of SARS-CoV-2 S-2P protein diluted to 5 μg/ml in PBS (pH 7.4) and incubated overnight at 4 °C. All subsequent incubations, until the addition of sucbstrate, were for 1-h at 37 °C. Wells were washed three times with PBS and incubated with blocking solution (5% (w/v) non-fat dried milk (NFDM)). Serial dilutions of human serum were prepared in

5% NFDM-PBS and added to the wells (25 μl per well) after the removal of the blocking solution. Plates were washed three times with PBS followed by the addition of secondary cross-adsorbed anti-human IgG-HRP (Thermo Fisher Scientific, Cat# 31413) detection antibody (25 μl per well) at 1:8000 dilution in 5% NFDM-PBS. Plates were washed three times with PBS. 1-Step Ultra TMB-ELISA Substrate Solution (Thermo Fisher Scientific, Cat# 34029) was added (25 μl per well) to detect binding, incubated at room temperature for 6–8-min followed by the addition of an equal volume of stop reagent (2 M sulfuric acid). Absorbance was measured at 450 nm using a Spectramax microplate Reader (Molecular Devices), and responses plotted as a function of dilution. Serum binding was calculated as the area under the curve using GraphPad Prism (version 9).

## IgM and IgG repertoire library construction

IgM libraries were prepared to determine the *IGHV* genotypes of the two study participants as previously described[53] *Frontiers* 2019). The IgM libraries were also used for Ab lineage tracing, and we therefore generated three IgM libraries from each participant. In brief, total RNA from ~5 million PBMCs from the acute infection timepoint was isolated using the Qiagen RNeasy mini kit. 400 ng RNA was reverse transcribed using Superscript IV (Invitrogen) and a gene-specific primer for the IgM constant region. For Ab lineage tracing, IgG libraries from different timepoints (acute infection, convalescent and post-vax) were prepared using a gene-specific primer for the IgG constant region as described (Vazquez Bernat et al. *Frontiers* 2019) and listed in Supplementary Data 3. For each library, paired-end reads were generated using the Illumina MiSeq 2 × 300 bp kit.

## Repertoire sequencing analysis

IgDiscover[25] was used to preprocess IgM/IgG libraries, annotate *VDJ* sequences, and infer individualized *V, D*, and *J* germline genotypes for the donors from the IgM libraries using the IMGT release 202226-1 (June 27, 2022) as a starting database. IgBLAST annotated Rep-seq *VDJ* sequences were denoised and deduplicated using Fast Amplicon Denoising (FAD) with an error rate of 0.0047, which is the median MiSeq error rate reported in ref. 54. We applied FAD[32] to reduce the prevalence of singleton sequences that differ by only one nucleotide, as these are more likely to represent different RNA templates from the same cell. This approach increases the likelihood that the traced sequences represent unique cells. Chimeric sequences were detected using a hidden Markov model (HMM) based approach. Briefly, the method generates an HMM based on an individual's personalized *IGHV* genotype. The V region of each query sequence is modeled as either being derived from a single *IGHV* allele with mutation or allowing template switching between different *IGHV* alleles (also with mutation, with mutation rates potentially differing between the different component templates). Comparing the two models' likelihoods provided us with a Bayes factor; we set a log10 Bayes factor of >5 as our threshold for chimera detection.

## Lineage tracing

Lineage tracing was performed on combined IgM Rep-seq, IgG Rep-seq, and mAb data with the IgDiscover clonotypes command[55]. A clonotype was defined as all sequences with the same *V* and *J* allele assignment, identical HCDR3 nucleotide lengths, and in the same single linkage cluster with a cutoff of 0.8 nucleotide match fraction between HCDR3 nucleotide sequences. Lineages with multiple mAbs which had differing light chain assignments were sub-split by first creating new clones for each light chain *VJ* combination, then reassigning the heavy chain Rep-seq sequences from the original clone to the sub-split lineage containing the mAb heavy chain sequence with the smallest possible Levenshtein distance. Clonotypes consisting of Rep-seq sequences and mAb sequences were given subdomain specificities and HCoV-HKU1 S binding data based on the mAbs they

contain. If any of the mAbs in a clone had different subdomain specificities or an undefined subdomain specificity, the clone was marked as having an undefined subdomain specificity.

## EC$_{50}$ calculation

Serum EC$_{50}$s were generated by fitting a logistic curve against the log$_{10}$ of the dilution factors using the nlsLM function from the minpack.lm R package. The logistic curve's maximum value was set to be the same across each antigen, with 1.364444 for anti-SARS2 S IgG and 1.818325 or anti-HCoV-HKU1 S IgG. These maximum values were computed as the median of per donor maximums across all readings for each antigen. Anti-SARS2 S IgG maximums below 0.5 were excluded from the median calculation because they were low outliers from the pre-pandemic timepoint. Signals that were too low and flat to fit a logistic curve were given an EC$_{50}$ equal to 100, which is the smallest dilution factor.

## Phylogenetic tree construction

All phylogenetic trees were created by first aligning nucleotide sequences using MAFFT v7.490[56] with default options, then generating maximum-likelihood trees using FastTree double precision v2.1.11 with the -gtr, -nt, -spr 4, -mlacc 2, and -slownni options for increased accuracy[57]. Visualization was performed using ggtree v3.2.1[58].

## Reporting summary

Further information on research design is available in the Nature Portfolio Reporting Summary linked to this article.

## Data availability

Repertoire sequencing data have been deposited to a restricted access repository at Science for Life Laboratory (SciLifeLab) Data Centre at https://doi.org/10.17044/scilifelab.21518142 to protect donor privacy. Data access requests may be submitted to the SciLifeLab Data Centre at datacentre@scilifelab.se with the DOI to establish a data-sharing agreement. The timeframe for a response to requests is less than a week. The starting database for IGH VDJ genotyping can be downloaded from IMGT V-Quest at https://www.imgt.org/download/V-QUEST/IMGT_V-QUEST_reference_directory/Homo_sapiens/IG/. Source data are provided with this paper.

## Code availability

IgDiscover22 v1.0.0 is available at https://gitlab.com/gkhlab/igdiscover22, scripts used to generate all results in the paper are available at https://gitlab.com/gkhlab/Vaccination_of_SARS-CoV-2_Convalescents and https://doi.org/10.5281/zenodo.7644243. HMM code for chimera identification is available at https://github.com/MurrellGroup/CHMMera/.

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

## Acknowledgements

This project was supported by a grant from the SciLifeLab National COVID-19 Research Program (C19VE:G:007) to M.N.E.F and G.K.H and funded by the Knut and Alice Wallenberg Foundation, as well as a Distinguished Professor grant from the Swedish Research Council (2017-00968) to GKH and a project grant (2020-06235) to M.N.E.F. We would like to thank the study participants and Claes Ahlm, Johan Norberg and the Clinical Research Center, Norrlands University Hospital, Umeå for acquisition of serum samples analyzed in Supplementary Fig. 1. We gratefully acknowledge the G2P-UK National Virology consortium funded by MRC/UKRI (grant ref: MR/W005611/1.) and the Barclay Lab at Imperial College for providing spike expression plasmids. Open access funding is provided by Karolinska Institutet. We also thank the Fondation Dormeur (Liechtenstein) for its generous contribution towards equipment.

## Author contributions

M.Ch., M.S., B.M., L.M.W., and G.B.K.H. designed the study, analyzed the results, and wrote the manuscript. M.S., R.I.C., H.L.D. C.G.R., and M.F. selected the samples. D.S. and P.F.W. performed neutralization studies and M.C. generated the Rep-seq libraries and performed the personalized genotyping. A.S. implemented the chimera detection method. All authors revised the manuscript and approved the final version prior to submission.

## Funding

## Competing interests

M.S., C.G.R., and H.L.D. are employees of Adimab LLC. and own equity stake in Adimab LLC. L.M.W. is an employee of Invivyd Inc. and owns shares in Invivyd Inc. The remaining authors declare no competing interests.
