## [Peer Review File · Nature Communications]

REVIEWER COMMENTS

Reviewer #1 (Remarks to the Author):

Several groups have reported that immunization previously SARS-CoV-2-infected subjects with mRNA-based vaccines results in a rapid expansion of the serum binding and neutralizing antibody responses, and of spike-specific memory B cell frequencies. In this study, Chernyshev and colleagues report that following infection and resolution of clinical symptoms, spike-specific B cells mature by accumulating somatic mutations in their BCRs and that upon immunization a diverse repertoire of B cell clones rapidly expands. These observations were made through a detailed characterization of an impressive number (459) of spike-specific monoclonal antibodies isolated from two infected+vaccinated subjects. Deep IGH repertoire sequencing analysis of these two subjects along with monoclonal antibody tracing at different time point following infection was also performed. Mabs were isolated both from plasma cells and from memory B cells. There are many, very interesting observations made in this study and despite the fact that B cell clonal analysis was performed in only two subjects, the study is experimentally solid and the results novel and important. I have no concerns.

The authors report that anti-HCoV-HKU1 infection significantly increases the titers of anti-HKU1 serum IgG titers and that the corresponding BCRs accumulate somatic mutation during infection.

-Did the anti-HKU1 neutralizing serum antibody titers also increase?

-Was the rate of somatic mutation accumulation in the anti-HCoV BCRs similar or different from that of anti-spike BCRs (during infection)?

Reviewer #2 (Remarks to the Author):

Another important contribution from Karlsson-Hedestam's group and collaborators, this study is a comprehensive analysis of antibody clonality, specificity and functionality in response to SARS-CoV-2 vaccination in convalescent subjects and informs the antibody repertoire of acute infection and vaccination, revealing how previous exposure to common coronavirus shapes vaccine response and binding to Omicron subvariants. In addition, at lines 341-349, the authors also did a good job of delineating the study limitations, i.e., questions that could arise while reading the paper but are not possible to be answered given lack of samples.

I have one minor comment and a suggestion that could be useful to improve the current version:

1) Sentence 73-38 is a bit confusing because it seems the authors are referring to acute infection only in this paragraph, but wrote "in contrast, at the acute infection...". I would rewrite for clarity, since it is also not clear by reading the sentence what is the hypothesis behind having more cross-reactive mAbs at the acute phase and why those are not maintained after vaccination.

2) I was curious to see whether there is an association between antibody features (either binding/functionality or V gene signatures) in mAbs found to bind pre-pandemic common cold coronaviruses and any of the variants? Meaning: are some antibodies that bind common cold coronaviruses and more frequently one of the variants?

Reviewer #3 (Remarks to the Author):

Review Chernyshev et.al. Nat Comm 2022

This is a clear, well written study describing the antibody response against the SARS-CoV-2 Spike protein during the COVID pandemic in individuals that were first infected and later received a vaccination. Although much has been published about the B cell response against SARS-CoV-2, also about this subject, it is still relevant to understand how B cell responses develop throughout infection and vaccination. Especially the longitudinal (lineage) tracing

is quite novel and informative although the final findings/conclusions are not very unexpected.

I have several issues which would be good to address prior to publication:

Major Issue

The authors may want to consider leaving out the first whole paragraph of the results section "Prior infection results in significantly increased antibody binding and neutralizing titers upon vaccination". All the data presented here is shown in supplementary files, has been described by others in more detail and it seems from the sampling profile in suppl Table 1 (which lacks donor codes) vs suppl Table 2 that the donors (IML3694 and IML3695) from which the antibodies and the bulk IgG libraries were retrieved are not part of the serum response data set (suppl Fig 1 vs Fig 2B)

Furthermore it is unclear why they have chosen to discover novel antibodies using the ACS isolation method for the acute, convalescent and post-vax samples but after the vaccination for a memory B cells sort using spike probes. How does it compare? In suppl Fig. 3 the distribution of IGHV is shown, since the ASC sorted cells were selected without any bias for S antigen recognition and MBC by using a S probe, there doesn't seem to be much difference between ASC and MBC, except for VH1-69.

The authors do mention in line 135 that they observe a VH1-69 peak post-vaccination. Many other studies have found VH1-69 to be dominant, also after convalescent infection. Can the authors explain why these were not found in the acute, convalescent and post-vaccination samples? Could that be due to the isolation method? In addition the IGHV3-53/66 gene segment is one of the most frequently used by mAbs isolated during the first wave - interesting to see that the authors also don't find back this gene.

Further the isolation of antibodies: how many mAbs were isolated from the ACS population and how many of those were S specific (absolute numbers or frequency). It is unclear how the antibodies from the ASC were selected after a 24 well culture of *S. cerevisiae* (ELISA?). In addition, can we assume that the majority of the spike sorted cells were indeed spike specific?

Since the authors combine the mAb data from ASC and MBC with the bulk IgG libraries it would be convenient if they explain how the bulk IgG library was generated.

Furthermore did the authors clone and reproduce data from the IgG library that were picked up with the lineage tracing; consider better mAbs with increased affinity, as well as low affinity clones ...

Minor Issue

Please check Suppl Fig4C, column 3 'timepoint' seems the IgM are all from the Acute timepoint

I think suppl Fig 1 is more informative compared to current Fig 1B which is interesting as well since it focuses on the two donors used throughout the manuscript

Use constant language to refer to the sample time point e.g. acute, convalescent, post-vax and pre-vax

Recently a paper from the same /collaborating group of Laura Walker (Sakharkar et al Sci Immunol 2021 and Kaku Sci Immunol 2022) showed that the immune response drift away from S2 towards RBD. This was found during break thru infection after vaccination, which is different from the situation described here, but maybe the authors can comment and discuss that since they seem to find more S2 specific antibodies (line 128/129).

Line 161/163: mg should be μg as indicated in Fig 2 which is also mislabeled in the figure legend.

Line 162: ADI-67444 is not in the study should be ADI-67744

Line 156/166: please have a careful look at the data of the VH3-53 and VH2-5. Doesn't seem that they are not exclusively RBD. Therefore the authors added almost , which is not very informative and not worth mentioning. The remark in Line 158 is not true for VH1-16 as determined in samples from the acute, convalescent and post-vax timepoint.

Line 201/202: The finding that 48% of the acute mAb were cross-reactive with HKU and thus suggest that the early response consisted of preexisting memory cells is interesting but the data as presented in suppl Table 3 makes it difficult to digest the mentioned remark.

In Fig 5C : in the figure C is not indicated

Please also check

**Wang, Z. et al. Analysis of memory B cells identifies conserved neutralizing epitopes on the N-terminal domain of variant SARS-Cov-2 spike proteins. Immunity (2022)
doi:10.1016/j.immuni.2022.04.003.**

Tong, P. et al. Memory B cell repertoire for recognition of evolving SARS-CoV-2 spike. Cell 184, 1–28 (2021).

Claireaux, M. et al. A public antibody class recognizes an S2 epitope exposed on open conformations of SARS-CoV-2 spike. Nat Commun 13, 4539 (2022).

Chen, E. C. et al. Convergent antibody responses to the SARS-CoV-2 spike protein in convalescent and vaccinated individuals. Cell Reports 36, 109604 (2021).

Robbiani, D. F. et al. Convergent antibody responses to SARS-CoV-2 in convalescent individuals. Nature 584, 437–442 (2020).

RESPONSE TO REVIEWER COMMENTS

Chernyshev et al. *Vaccination of SARS-CoV-2-infected individuals expands a broad range of clonally diverse affinity-matured B cell lineages*, NCOMMS-22-44454

Reviewer #1

Several groups have reported that immunization previously SARS-CoV-2-infected subjects with mRNA-based vaccines results in a rapid expansion of the serum binding and neutralizing antibody responses, and of spike-specific memory B cell frequencies. In this study, Chernyshev and colleagues report that following infection and resolution of clinical symptoms, spike-specific B cells mature by accumulating somatic mutations in their BCRs and that upon immunization a diverse repertoire of B cell clones rapidly expands. These observations were made through a detailed characterization of an impressive number (459) of spike-specific monoclonal antibodies isolated from two infected +vaccinated subjects. Deep IGH repertoire sequencing analysis of these two subjects along with monoclonal antibody tracing at different time point following infection was also performed. Mabs were isolated both from plasma cells and from memory B cells. There are many, very interesting observations made in this study and despite the fact that B cell clonal analysis was performed in only two subjects, the study is experimentally solid and the results novel and important. I have no concerns. The authors report that anti-HCoV-HKU1 infection significantly increases the titers of anti-HKU1 serum IgG titers and that the corresponding BCRs accumulate somatic mutation during infection.

-Did the anti-HKU1 neutralizing serum antibody titers also increased?

We thank the Reviewer for the positive comments about our study, and we appreciate this question. We do not have the technical capabilities established to assess serum neutralizing activity against HKU1. There are now many papers showing that SARS-CoV-2 infection boosts serum binding responses to seasonal beta CoVs, but neutralizing activity was not assessed in those studies. Authentic virus neutralization assays for HKU1 and OC43 are not simple to set up and we are unable to do so for the current study. However, we note that the HKU1 spike cross-reactive monoclonal antibodies isolated in our study rarely bind the RBD subunit, the primary target for neutralizing antibodies. Therefore, while not a definitive answer, we do not expect that a primary SARS-CoV-2 infection appreciably boosts serum neutralizing activity against HKU1. To support this, we have added a panel as Supplementary Figure 2C, which shows the sub-specificities of the different HKU1 cross-reactive mAbs isolated at the different time points and inserted a comment describing those results on lines 127-128 of the manuscript.

-Was the rate of somatic mutation accumulation in the anti-HCoV BCRs similar or different from that of anti-spike BCRs (during infection).

Figure 3B shows the SHM of the HKU1 S-specific mAbs isolated at the acute infection time point compared to the SARS-CoV-2 S-specific mAbs isolated at the same time point. We cannot generate SHM line-plots for the HKU1 S-binding Abs since there are too few sequences detected for a given lineage across time points. However, the average SHM values for HKU1 S cross-reactive sequences that could be traced at both pre- and post-vax time points were not greater than those observed for the HKU1 S cross-reactive sequences at the acute infection time point.

Reviewer #2

Another important contribution from Karlsson-Hedestam's group and collaborators, this study is a comprehensive analysis of antibody clonality, specificity and functionality in response to SARS-CoV-2 vaccination in convalescent subjects and informs the antibody repertoire of acute infection and vaccination, revealing how previous exposure to common coronavirus shapes vaccine response and binding to Omicron subvariants. In addition, at lines 341-349, the authors also did a good job of delineating the study limitations, i.e., questions that could arise while reading the paper but are not possible to be answered given lack of samples.

I have one minor comment and a suggestion that could be useful to improve the current version: 1) Sentence 73-38 is a bit confuse because it seems the authors are referring to acute infection only in this paragraph, but wrote "in contrast, at the acute infection...". I would rewrite for clarity, since is also not clear by reading the sentence what is the hypothesis behind having more cross-reactive mAbs at the acute phase and why those are not maintained after vaccination.

We thank the Reviewer for the positive comments about our study and for bringing up this valid point, which we have also discussed ourselves. The presence of cross-reactive HKU1 and SARS-CoV-2 spike mAbs suggests that a common epitope is exposed during infection. However, if HKU1-cross-reactive mAb lineages are not, or only rarely, boosted by the vaccination, it suggests that they target an epitope(s) that is not present on the pre-fusion stabilized spike used in the vaccine, such as those present on the post-fusion conformation of spike. Indeed, the commercial HKU1 spike protein used in our binding assays was not in a pre-fusion stabilized form.

There could also be an immunological reason as follows. During the early acute SARS-CoV-2 infection of previously unexposed individuals, circulating memory B cells elicited by prior seasonal beta-CoV infections are the only circulating memory B cells that recognize the SARS-CoV-2 spike and they are therefore preferentially activated to become plasmablasts over the lower affinity, lower frequency SARS-CoV2-specific naive B cells present at the same time. In parallel, new SARS-CoV-2-specific naive B cells are recruited to the GC where they affinity mature and exit as memory B cells. When vaccination occurs several months later, the better matched SARS-CoV-2 spike-driven memory B cell repertoire is preferentially boosted. At this point the HKU1 cross-reactive memory B cells have lost the advantage they had during the acute infection phase.

The Reviewer refers to line 73-38 (we assume the original lines 73-88 are intended), which previously read:

"Of the 459 spike-binding mAbs, a set of mAbs (n=33) bound both the SARS-CoV-2 and the HCoV-HKU1 spike. The cross-reactive mAbs were found predominantly at the acute infection time point and likely originated from pre-existing MBCs as they displayed significant levels of somatic hypermutation (SHM) already at this time point. In contrast, at the acute infection time point the SARS-CoV-2 S-specific mAbs had low SHM and many of the lineages could be traced to the IgM repertoire, consistent with de novo elicitation."

We have now changed this to:

“Of the 459 spike-binding mAbs, a set of mAbs (n=33) bound both the SARS-CoV-2 and the HCoV-HKU1 spike. The cross-reactive mAbs were found predominantly at the acute infection time point and likely originated from pre-existing MBCs generated by a prior infection with HKU1 or a related beta-CoV. The HKU1-cross-reactive mAbs displayed significantly higher levels of somatic hypermutation (SHM) at the acute infection time point than the SARS-CoV-2 S-specific mAbs isolated from the same time point. Furthermore, except for a single sequence, the HKU1 cross-reactive lineages could not be traced in IgM repertoires from the acute SARS-CoV-2 infection time point, unlike the SARS-CoV-2 S-specific lineages, which could be traced in the IgM repertoire, consistent with *de novo* elicitation of the latter.”

2) I was curious to see whether there is an association between antibody features (either binding/functionality or V gene signatures) in mAbs found to bind pre-pandemic common cold coronaviruses and any of the variants? Meaning: are some antibodies that bind common cold coronaviruses and more frequently one of the variants?

No, none of the SARS-CoV-2 variant neutralizing antibodies bound HKU1 S in our study.

Reviewer #3

This is a clear, well written study describing the antibody response against the SARS-CoV-2 Spike protein during the COVID pandemic in individuals that were first infected and later received a vaccination. Although much has been published about the B cell response against SARS-CoV-2, also about this subject, it is still relevant to understand how B cell responses develop throughout infection and vaccination. Especially the longitudinal (lineage) tracing is quite novel and informative although the final findings/conclusions are not very unexpected. I have several issues, which would be good to address prior to publication:

Major Issue

The authors may want to consider leaving out the first whole paragraph of the results section “Prior infection results in significantly increased antibody binding and neutralizing titers upon vaccination”. All the data presented here is shown in supplementary files, has been described by others in more detail and it seems from the sampling profile in suppl Table 1 (which lacks donor codes) vs suppl Table 2 that the donors (IML3694 and IML3695) from which the antibodies and the bulk IgG libraries were retrieved are not part of the serum response data set (suppl Fig 1 vs Fig 2B)

We thank the Reviewer for the positive comments about our study. We understand the concerns about Supplementary Figure 1 given the many previous reports on this topic, and we have now shortened this section considerably. However, we do feel that it should not be entirely removed as these data provide a context to the questions addressed in the current study, especially since we focused on only two subjects for the in-depth longitudinal Ab lineage characterization. Since the other two reviewers did not comment on this, we feel that it is most fair to leave this section in the paper, although in a more concise format.

IML3694 and IML3695 are part of the serum response dataset but were not in Supplementary Table 1. We apologize for the confusion. Supplementary Table 1 now has donor codes and includes IML3694/IML3695. Supplementary Table 2 would then include duplicate information, so we have removed it and we reference Supplementary Table 1 instead in the revised version of the manuscript.

Furthermore, it is unclear why they have chosen to discover novel antibodies using the ACS isolation method for the acute, convalescent, and post-vax samples but after the vaccination for a memory B cells sort using spike probes. How does it compare? In suppl Fig. 3 the distribution of IGHV is shown, since the ASC sorted cells were selected without any bias for S antigen recognition and MBC by using a S probe, there doesn't seem to be much difference between ASC and MCB, except for VH1-69.

We thank the Reviewer for these questions and would like to clarify that even though the mAbs isolated from ASCs were not pulled out with a spike probe, they were tested for spike binding after expression and only antibodies that bound the SARS-CoV-2 spike trimer (either binding S-2P or a non-stabilized S protein) were included in the study. As pointed out, the IGHV gene usage was very similar across the time points and isolation methods, and largely reflects the over IGHV gene usage in the total IgG repertoire as shown in Supplemental Figure 4D.

The authors do mention in line 135 that they observe a VH1-69 peak post-vaccination. Many other studies have found VH1-69 to be dominant, also after convalescent infection. Can the authors explain why these were not found in the acute, convalescent, and post-vaccination samples? Could that be due to the isolation method?

Even though the proportion of IGHV1-69-using antibodies over the course of infection for the two donors is on the lower end of the range observed in other studies, we did isolate numerous IGHV1-69-using antibodies from all time points, with the post-vaccination time point yielding a particularly large number of such antibodies. This could be because some of the IGHV1-69 Abs recognize a previously described epitope in S2 that becomes exposed on S proteins used in approved vaccines (described by Claireaux et al. in: A public antibody class recognizes an S2 epitope exposed on open conformations of SARS-CoV-2 spike - PubMed (nih.gov)) that use IGHV1-69/IGKV3-11. We found 13 post-vax IGHV1-69/IGKV3-11 pairings, 7 in IML3694 and 6 in IML3695, of which 12 were S2-directed. These data are included in Supplementary Table 3 and a comment to this effect has now been inserted in the manuscript (line 136-137).

In addition, the IGHV3-53/66 gene segment is one of the most frequently used by mAbs isolated during the first wave - interesting to see that the authors also don't find back this gene.

We agree with the Reviewer that we isolated relatively few IGHV3-53-using mAbs in the current study. However, we think that the two donors analyzed here fall within the range typically observed when isolating SARS-CoV-2 spike-binding mAbs from different donors. Across several studies, the IGHV genes that are most frequently used in the spike-specific response are IGHV3-30 and IGHV3-30-3, usually followed by IGHV3-33 and IGHV1-69, then IGHV4-31 and IGHV5-51 and only after this IGHV3-53 and IGHV3-21 and other genes (see for example Sakharkar et al. *Science Immunology* 2021 and Pushparaj et al. *Immunity* 2022).

Some published studies report a greater proportion of IGHV3-53-using mAbs, but this is usually following B cell sorting using an RBD probe, which enriches for such Abs (Rogers et al. *Science* 2020, Gaebler et al. *Nature* 2021, Zhou et al. *Cell Reports* 2021).

Further the isolation of antibodies: how many mAbs were isolated from the ACS population and how many of those were S specific (absolute numbers or frequency). It is unclear how the antibodies from the ASC were selected after a 24 well culture of *S. cerevisiae* (ELISA?). In addition, can we assume that the majority of the spike sorted cells were indeed spike specific?

Depending on donor and timepoint, the efficiency of isolating and cloning spike-specific ASCs was between 9-46%. The overall efficiency is subject to attrition by any limitations in the yeast-cloning

platform. Alternatively, for MBCs, between 30-47% of isolated antibodies were binders. Since ASCs have limited surface BCR expression, we did not want to restrict the/bias the sorting of B cells activated by the recent acute infection. Importantly, all 459 mAbs included in the study, whether sorted from ACSs (without probe) or memory B cells (with probe), were confirmed to bind the SARS-CoV-2 spike trimer after they were expressed.

Since the authors combine the mAb data from ASC and MBC with the bulk IgG libraries it would be convenient if they explain how the bulk IgG library was generated.

We apologize for this omission. The IgG libraries were generated according to the protocol described in Vazquez Bernat et al. *Frontiers in Immunology* 2019. We have now added a section outlining this in the **Methods** section.

Furthermore, did the authors clone and reproduce data from the IgG library that were picked up with the lineage tracing; consider better mAbs with increased affinity, as well as low affinity clones ...

We appreciate this question as we performed such experiments in a previous study (Phad et al. *JEM* 2020). In the current study, we opted not to do so as the focus of this study was to understand how the overall polyclonal (infection-induced) spike-specific B cell repertoire evolves longitudinally before and after vaccination rather than to delve into interesting Ab specificities and improved somatic relatives of these. As the Reviewer indicates, the novelty with our study is the fact that we couple mAb isolation with lineage tracing in bulk repertoire NGS data to identify as many somatic variants of each mAb lineage as possible. The purpose was to allow a deeper analysis of the response compared to other approaches reported before, such as the isolation of mAbs at different time points from the same individual, where only the most expanded lineages tend to be picked up several times. In addition, we believe the value of pursuing mAbs with increased affinity may be limited by the fact that these donors were infected with an early strain of SARS-CoV-2.

Minor Issue

Please check Suppl Fig4C, column 3 'timepoint' seems the IgM are all from the Acute timepoint

That is correct, we generated multiple IgM libraries from the acute time point from each donor. We have now clarified this in the **Methods** section describing library preparation.

I think suppl Fig 1 is more informative compared to current Fig 1B which is interesting as well since it focuses on the two donors used throughout the manuscript

The Reviewer is correct in that Figure 1B shows the response in the two donors that were followed throughout the manuscript. In contrast, Supplementary Figure 1 shows a larger cohort to provide a context. In response to this Reviewer's suggestion that we remove or shorten the description of the results shown in Supplementary Figure 1, we have now done so. However, we believe that it is important to keep the Figure, along with a brief description.

Use constant language to refer to the sample time point e.g. acute, convalescent, post-vax and pre-vax

This has been corrected.

Recently a paper from the same /collaborating group of Laura Walker (Sakharkar et al *Sci Immunol* 2021 and Kaku *Sci Immunol* 2022) showed that the immune response drift away from S2 towards RBD. This was found during break thru infection after vaccination, which is different from the situation

described here, but maybe the authors can comment and discuss that since they seem to find more S2 specific antibodies (line 128/129).

We thank the reviewer for this suggestion. The Kaku et al 2022 study evaluated the early antibody response following BA.1 breakthrough infection in mRNA vaccinated (never previously infected) individuals. These results showed that, despite the relative conservation of the BA.1 S2 subunit compared with the RBD, BA.1 breakthrough infection preferentially boosted cross-reactive antibodies targeting the RBD. As noted in the Kaku et al manuscript, the molecular explanation(s) for the dampened antibody response to the S2 subunit remain to be determined but may be driven by the increased serum antibody masking of the conserved S2 subunit relative to the more divergent RBD, resulting in a limited S2 epitope accessibility for B cell targeting. Conversely, the extensive immune evasion of the BA.1 RBD may have resulted in substantially lower levels of serum antibody feedback, potentially enabling the activation of rare cross-reactive RBD-directed MBCs, similar to a recent study from the Nussenzweig group Increased memory B cell potency and breadth after a SARS-CoV-2 mRNA boost - PubMed (nih.gov). In contrast, as noted by the reviewer, in our current study of ancestral SARS-CoV-2 infected/vaccinated individuals, we observed a high frequency of antibodies to the S2 subunit at both early and late time points following infection and following vaccination. Although the factors driving the different patterns of immunodominance in these two exposure settings remain to be elucidated, the results suggest that the type (homologous vs heterologous exposure), route (systemic vs mucosal), and/or interval between exposures likely influences B cell immunodominance hierarchy. We have now commented on this in the text in lines 318-324.

Line 161/163: mg should be μg as indicated in Fig 2 which is also mislabeled in the figure legend.

This has been corrected.

Line 162: ADI-67444 is not in the study should be ADI-67744

This has been corrected.

Line 156/166: please have a careful look at the data of the VH3-53 and VH2-5. Doesn't seem that they are not exclusively RBD. Therefore, the authors added almost, which is not very informative and not worth mentioning. The remark in Line 158 is not true for VH1-16 as determined in samples from the acute, convalescent, and post-vax timepoint.

We have removed: "except IGHV3-53 and IGHV2-5, which were almost exclusively used in the Ab response to RBD (Fig. 1E).

Line 201/202: The finding that 48% of the acute mAb were cross-reactive with HKU and thus suggest that the early response consisted of preexisting memory cells is interesting, but the data as presented in suppl Table 3 makes it difficult to digest the mentioned remark.

We believe that the reviewer misread the text, there is no mention of 48% on lines 201/202. The 48% is referenced on line 190 and refers to the fact that 48% of all HKU1 cross-reactive mAbs were obtained from the acute timepoint. 16 of 66 (24%) acute timepoint mAbs are cross-reactive with HKU1. The 48% has also been rounded up to 49%.

In Fig 5C: in the figure C is not indicated

We apologize and thank the Reviewer for noticing this. A "C" has now been added.

Please also check Wang, Z. et al. Analysis of memory B cells identifies conserved neutralizing epitopes on the N-terminal domain of variant SARS-Cov-2 spike proteins. Immunity (2022)

doi:10.1016/j.immuni.2022.04.003.

Tong, P. et al. Memory B cell repertoire for recognition of evolving SARS-CoV-2 spike. *Cell* 184, 1–28 (2021).

Claireaux, M. et al. A public antibody class recognizes an S2 epitope exposed on open conformations of SARS-CoV-2 spike. *Nat Commun* 13, 4539 (2022).

Chen, E. C. et al. Convergent antibody responses to the SARS-CoV-2 spike protein in convalescent and vaccinated individuals. *Cell Reports* 36, 109604 (2021).

Robbiani, D. F. et al. Convergent antibody responses to SARS-CoV-2 in convalescent individuals. *Nature* 584, 437–442 (2020).

We thank the Reviewer for this and have inserted references to Robbiani et al. (line 155), Wang et al. (line 294), and Claireaux et al. (line 134) in the relevant sections.